# Antibacterial Application on *Staphylococcus aureus* Using Antibiotic Agent/Zinc Oxide Nanorod Arrays/Polyethylethylketone Composite Samples

**DOI:** 10.3390/nano9050713

**Published:** 2019-05-08

**Authors:** Dave W. Chen, Kuan-Yi Lee, Min-Hua Tsai, Tung-Yi Lin, Chien-Hao Chen, Kong-Wei Cheng

**Affiliations:** 1Department of Orthopaedic Surgery, Chang Gung Memorial Hospital, Keelung Branch, Taoyuan 204, Taiwan; m0423042@stmail.cgu.edu.tw (K.-Y.L.); m0523048@stmail.cgu.edu.tw (M.-H.T.); ross_1222@hotmail.com (T.-Y.L.); chchen1982@gmail.com (C.-H.C.); 2College of Medicine, Chang Gung University, Taoyuan 333, Taiwan; 3Department of Chemical and Materials Engineering, Chang Gung University, Taoyuan 333, Taiwan

**Keywords:** antibacterial properties, zinc oxide, nanorods, polyetheretherketone (PEEK)

## Abstract

In this study, zinc oxide (ZnO) nanorod arrays as antibiotic agent carriers were grown on polyetheretherketone (PEEK) substrates using a chemical synthesis method. With the concentration of ammonium hydroxide in the precursor solution kept at 4 M, ZnO nanorod arrays with diameters in the range of 100–400 nm and a loading density of 1.7 mg/cm^2^ were grown onto the PEEK substrates. Their drug release profiles and the antibacterial properties of the antibiotic agent/ZnO/PEEK samples in the buffer solution were investigated. The results showed that the concentrations of antibiotic agents (ampicillin or vancomycin) released from the samples into the buffer solution were higher than the value of minimum inhibitory concentration of 90% for *Staphylococcus aureus* within the 96 h test. The bioactivities of ampicillin and vancomycin on substrates also showed around 40% and 80% on the *Staphylococcus aureus*, respectively. In the antibacterial activity test, sample with the suitable loading amount of antibiotic agent had a good inhibitory effect on the growth of *Staphylococcus aureus*.

## 1. Introduction

Applications of biomaterials with suitable antibaterial properties have been employed for the treatments of many diseases and clinical conditions that are incurable today. In orthopaedic applications, the biomaterials with suitable infection-prevention and promoting bone-healing abilities are major research topics. Biomaterials in orthopedics such as titanium-based alloys, cobalt-based alloys, or polyetheretherketone (PEEK) are often used for the treatments in complex musculoskeletal wounds [1]. However, poor antibacterial performances for these biomaterials influence the recovery time of postoperative care. Therefore, the possible surface modifications for these biomaterials that can be applied in orthopedic suffers have to be developed in order to meet clinical requirements [1,2]. Examples of natural surfaces with antibacterial properties include plant leaves, gecko feet, shark skin, insect wings, fish scales, and spider silk [2,3,4]. Interesting nanostructures with antibacterial properties such as nanorod arrays can be found in the Cicada wing surfaces [4]. The cells that attach to the Cicada wings are mechanically ruptured with the particular surface nanorod microstructures in a short period of time. The development of similar nanostructures in biomaterials with suitable antibacterial properties was reported by Pogodin et al. [5]. The results proposed by Pogodin et al. [5] give us a new concept of the surface modification of biomaterials. Zinc oxide (ZnO) is a traditional metal oxide that is non-toxic with a high melting point, good thermal stability,, low manufacturing costs, and good antibacterial property [6,7,8]. Various methods such as chemical vapor deposition [9], magnetic sputtering [10], hydrothermal reaction [11], and the template method [12] have been employed for the preparation of metal oxide thin films such as ZnO or TiO_2_ with 2D microstructures for the applications of optical-electronic devices [12], catalysts [6], photo-active layers in the solar cells [5], and the improvement of the sample’s antibacterial properties [13,14,15,16,17,18,19,20,21,22,23]. Ipeksac et al. [14] prepared the hydrothermal synthesized ZnO nanotubes in solution bath at a low temperature with the ZnO seeds coated onto the nickel filter. Their composite ZnO nanotubes/nickel filter samples showed the good antibacterial performances on the *Staphylococcus aureus* (*S. aureus*) bacteria. Roguska et al. [15] prepared the TiO_2_ nanotube arrays of Ti substrates using electrochemical anodization in a solution bath containing NH_4_F_,_ deionized water, and glycerol. Silver and ZnO nanoparticles were then grown onto the Ti samples using the sputtering and electrodeposition, respectively. The release of the Ag^+^ ions from the silver powders and Zn^2+^ ions from ZnO nanoparticles attached on the TiO_2_ samples could inhibit the growth of Gram-positive and negative bacteria. Zhang et al. [17] also tested the antibacterial performance and the osteoinductivity of the ZnO and strontium (Sr) materials coated on the surfaces of TiO_2_ nanotubes. The antibacterial properties of around 100% on the *S. aureus* and *Escherichia coli* (*E. coli*) growth using their samples were observed compared with almost 0% antibacterial property for pure TiO_2_ sample. Because the good antibacterial performances of ZnO samples were reported in the literature, the ZnO sample may have good application in the treatment of complex musculoskeletal wounds. Then, we examined reports about the mechanisms of the antibacterial properties for ZnO samples in the literature [23,24,25,26,27,28]. The antibacterial property of ZnO sample is due to (a) the formation of reactive oxygen spices (ROS) such as OH^−^ or O_2_^2−^ ions at the basic solution [26,28], (b) the release of Zn^2+^ ions from the ZnO samples [24,28], and/or (c) the ROS ions generated under light irradiation [25]. With the application of biomaterials in the human body, it is difficult to generate the ROS ions using the ZnO sample under light irradiation. Singh et al. [28] also shows that the formation of ROS ions and the release of Zn^2+^ ions from the ZnO sample are easier in the basic solution (pH value of greater 14) than those in the basic solution with low pH value (pH value = 12) or the buffer solution (pH of around 7). Regarding the application of biomaterials in the human body, poor antibacterial properties of the ZnO thin film coated on the biomaterials is thus expected due to the lacks of light irradiation and the nearly neutral condition of the human body. The release of Zn^2+^ ions from the ZnO sample into the human body is also unacceptable. Therefore, the surface morphology of the ZnO sample has to be remodified in order to obtain good antibacterial properties and good attachments of bone cells. Our previous study showed that ampicillin sodium salt and vancomycin hydrochloride can be directly absorbed onto the surface of polylactic acid (PLA) due to it having many hydroxyl groups [29]. However, the PLA is a biodegradation polymer that cannot be used as a bone support. For orthopedic applications, metal-based biomaterials such as titanium alloy or stainless steels are often used in the clinical treatments. The possibility of metal ion release caused by corrosion and mismatched elastic moduli between these metals (110 GPa for titanium alloy) and human bone (18 GPa) has always caused the failure of metal-based implants [30]. PEEK is a semi-crystalline polymer with several excellent properties such as hard tissue implant material, good elastic modulus (3–4 GPa), and good thermal and chemical properties that can avoid the degradation of implants caused by corrosion [30]. It indicates that the PEEK material is a good implant candidate for orthopedic applications [30]. However, few hydroxyl groups, carboxyl groups, and amines on the PEEK surface may result from the difficult attachments of bone cells or antibiotic agents onto the PEEK surface. Therefore, the development of possible methods for obtaining ZnO nanostructures with controlled properties, such as shape or size on PEEK substrates, is also important for potential optical, electronic, and antibiotic applications. At present, the synthesis of nanomaterials with the control properties of these obtained nanostructures—repeatability and reproducibility—is a new research topic [31,32,33]. Wojnarowicz et al. [31] discussed the size control mechanism of ZnO nanoparticles using the microwave solvothermal synthesis method. Their results showed the size distributions of ZnO particles in the range of 20–120 nm through the control of water content in the solution of zinc acetate in ethylene glycol bath. Lee and Leem [33] also prepared the ZnO nanorod arrays using a hydrothermal synthesis with various concentrations of chemicals in precursor solutions. Their results showed that the optical and other properties of ZnO nanorods can be controlled with a suitable size distribution using the control of concentrations for chemicals in a precursor solution. Although the synthesis of ZnO nanostructures on various substrates has been reported in the literature, few reports have discussed the growth of ZnO nanorod arrays onto the PEEK substrate. If we can deposit the ZnO nanorod arrays onto the surface of a PEEK sample, the hydroxyl groups will be easily generated at the ZnO sample surface when the ZnO nanorods make contact with the buffer solution. This could also provide suitable surface areas for the loading of antibiotic agents on the ZnO samples. Growth of ZnO nanorod arrays and the loading of the antibiotic agents on samples therefore improve both the antibacterial property and the attachments of bone cells onto the PEEK sample surfaces. Recently, 3D printing technology has become an interesting and fast production technology that can easily prepare implants for orthopaedic sufferers. Using a suitable design for the product, a 3D object with layer-by-layer building using a 3D printer can be obtained in several hours. The applications of 3D printing have many benefits such as un-moldable printing, an infinite variety of shapes for printing, rapid printing for digital design, and printing at point of care. An important benefit in the treatment of complex musculoskeletal wounds for the orthopedic sufferer is the rapid printing and infinite variety of shapes of the printing [29]. Therefore, we prepared the ZnO nanorod arrays on the 3D printing PEEK substrates using the simple chemical synthesis method. The antibiotic agent was directly absorbed onto the surface of ZnO nanorod arrays in the solution bath containing the antibiotic agent with various concentrations. In-vitro antibiotic agent release rates from the sample surfaces into the buffer solutions were analyzed using high-performance liquid chromatography (HPLC). The bacterial inhibition tests and measurement of optical densities of the *S. aureus* in the nutrient broth (NB) solutions were also carried out to evaluate the antibacterial properties of the ZnO/PEEK samples with and without the loading of antibiotic agent.

## 2. Materials and Methods

In this study, we tried to deposit the ZnO nanorod arrays onto the 3D printing PEEK substrates using the chemical synthesis method. Detail apparatus and procedures for the preparation of PEEK disks using 3D printer were similar to those reported in our previous study [29]. However, the melting and glass transition points for the PEEK are higher than those for the PLA sample. The temperature of nozzle in the 3D printer was changed from 220 °C to 340 °C in order to meet the requirements of thermal fused 3D printing technology. The temperature of holder for the 3D printing PEEK sample was kept at 45 °C. The printing speed was kept at 5 mm/s. In order to grow the ZnO nanorods on the PEEK substrates, an activation process of the substrate has to be carried out [34]. A solution containing 20 mL of 0.5 mM potassium permanganate (KMnO_4_, purity > 99%, Aldrich Co., Darmstadt, Germany) and 50 μL of 1-butanol (CH_3_(CH_2_)_3_OH, purity > 99%, Alfa Aesar Co., Lancashire, United Kingdom) was used for the activation of substrates. According to the results proposed by Kokotov and Hodes [34], the addition of potassium permanganate with the existence of 1-butanol as the reduction agent in solution bath can make the growth of the Mn–hydroxyoxide on the substrate. It can act as the effective seed layer that contributes the formation of ZnO nanorods on the substrate in the alkaline solution. According to the results proposed by Kokotov and Hodes [34], the concentration of ammonium hydroxide in the precursor solution is an important factor that influences the microstructures of ZnO nanorod arrays on glass substrates. High concentration of ammonium hydroxide in the solution results in the high pH value for the solution, which makes the decomposition of ZnO nanorod arrays occur. Low ammonium hydroxide concentration in the reaction bath results in the low growth rate of ZnO samples [34,35]. In order to obtain the uniform distribution of seed layer on substrate, the PEEK samples were put into the above solution and maintained under ultra-sonication bath with a temperature of 85 °C for 20 min. After the activation process, the substrates were maintained in the water bath under ultra-sonication bath for 10 min.

For the growth of ZnO nanorod arrays on substrates, a reaction solution containing 2 mL of 1 M zinc nitrate (Zn(NO_3_)_2_ 6H_2_O, purity > 98%, Sigma-Aldrich Co., Darmstadt, Germany), 3 mL with various concentrations (4.0–5.5 M) for ammonium hydroxide (NH_4_OH, J. T. Baker Co., Leicestershire, United Kingdom), 2 mL of ethanolamine (MEA, NH_2_CH_2_CH_2_OH, purity > 99%, Riedel-de Haën Co., Shanghai, China), and 13 mL of deionized water was prepared and maintained in the glass container. The substrates after the activation process were directly put into the reaction solution and maintained in the oil bath with the reaction temperature kept at 85 °C for 40 min in order to obtain the ZnO nanorod arrays grown on the substrates. After the growth of ZnO nanorod arrays on substrates, the samples were dried in an oven with temperature of 70 °C for 30 min. In order to get good attachments of ZnO nanorod arrays on the PEEK substrates, an annealing process of samples has to be carried out. Annealing of ZnO composite samples can also improve their chemical resistance abilities for corrosion in solution bath. However, the high temperature and long- annealing time caused the damage of PEEK substrates. Therefore, the as-prepared samples were annealed in the rapid thermal annealing system (ULVAC-RIKO, RHL-P610CP, Kanagawa, Japan) with a temperature increasing rate of 100 °C/min and maintained at 330 °C for 10 min.

The crystal phase, surface morphology, composition, and effective surface area of the ZnO samples on substrates were examined using the X-ray diffractometer (XRD, D2 phaser, A26-X1-A2B0B2A, Bruker Co., New Taipei City, Taiwan) with CuKα (λ = 1.5418 Å) irradiation, a filed-emission scanning electron microscope (FE-SEM, JEOL JSM-7500F), and a scanning electron microscope (S-3000N, Hitachi, Tokyo, Japan) equipped with the energy dispersive spectrometer (EDS, HORIBA, 7021-H) with the acceleration voltage of 15 kV and working distance of 15 mm, and a specific surface area analyzer (Micomeritics, ASAP 2020, Norcross, GA, USA). For the measurement of specific surface area of a sample, a degassing process has to be carried out in order to remove the gases that may have physically absorbed onto the sample surface. For the degassing process, the sample (of around 0.3 g) was put in a container loaded in the specific surface analyzer. The temperature and pressure of the container were kept at 90 °C and 3 μm-Hg in order to remove the gases physically absorbed on the sample, respectively. Total degassing process time was kept at 1000 min in order to obtain a clean sample. After the degassing process, total weight of the clean sample without any gas absorbed on the sample can be obtained using the weighting method. The gas for the analysis of sample’s surface area is the nitrogen gas with the temperature kept at 77.4 K (the boiling point of nitrogen). The special surface area of sample was calculated using the nitrogen adsorption method based on the linear form of the BET (Brunauer-Emmett-Teller) isotherm equation with the absorption range of P/Po kept in 0.06–0.6.

For the preparation of antibiotic agents on the ZnO samples, two antibiotic agents were tested in the study. Ampicillin sodium salt (C_16_H_18_N_3_NaO_4_S, purity > 98%) and the vancomycin hydrochloride (C_66_H_75_Cl_2_N_9_O_24_ HCl, purity > 98%) were provided from the Aldrich Co, Lancashire, United Kingdom. The organism used for the bacterial inhibition test was the *S. aureus* (ATCC6538R), provided by the Bioresource Collection and Research Center (BCRC, Hsinchu, Taiwan). The NB (beef extract 3%, peptone 5 g) was used for the test of bioactivity of *S. aureus*. The approach for the preparation of ZnO/antibiotic agent sample on PEEK substrates was the direct absorption of antibiotic agent onto the ZnO sample surface. Various concentrations of antibiotic agents (5, 10, and 15 mg/mL) in water baths were used for the preparation of the ZnO/antibiotic agent samples. The ZnO/PEEK sample with an average area of around 1 cm^2^ and a ZnO loading density of 1.7 mg/cm^2^ was placed in the aqueous solution containing suitable concentration of antibiotic agent at the room temperature for two days in dark conditions in order to avoid any influence of light during the absorption. After the full absorption of antibiotic agent on the sample surface, the samples were kept in a clean container in order to avoid any possible influence from other chemicals or organisms.

In vitro elution test was employed for the determination of drug release rate from the samples. Similar approach was reported in our previous study [29]. The following is a brief description. The phosphate buffer solution (pH 7.4, 25 mL, Sigma-Aldrich Co.) was used for the test of drug release behavior from the sample at a temperature of 37 °C with a wavering rate of 30 rpm. In vitro elution test was analyzed within the time interval of 1 h using the HPLC (JASCO Co., Pu-2080, Tokyo, Japan) with the SYMMETRY C_8_ column (4.6 × 250 mm, Shim-pack, VP-ODS, Tokyo, Japan). The phosphate buffer solution (25 mL) was replaced every 1 h in order to avoid the influence of saturated concentration for antibiotic agent in the buffer solution. The mobile phase, absorbency, and flow rate of mobile phase were the same as our previous study [29]. The calibration curves for the antibiotic agents in the water bath and phosphate buffer solution (both correction coefficient > 0.99) were made for the determination of unknown concentration of the antibiotic agent in the aqueous or buffer solution. The concentrations of Zn^2+^ions released from the ZnO samples were also analyzed using inductively couple plasma optical emission spectrometry (Varian, Vista-Pro ICP-OES, Corona, CA, USA) in order to examine the influence of cell toxicity of our samples.

The method for the test is the same as our previous study [29]. 200 μL *S. aureus* inoculum was cultured in 5 mL NB solution and grown for 12 h at 37 °C with a constant shaking rate of 200 rpm. Finally, the concentration of bacterial suspension was adjusted to around 10^8^ colony-forming unit (CFU)/mL. The antibiotic disk diffusion method for *S. aureus* in the agar containing NB was carried out in the Petri dish with sample in order to evaluate its bioactivity property. 200 μl solution containing organisms with the bacterial concentration of 10^8^ CFU/mL was pipetted and seeded onto the agars in the Petri disks for the test. The inhibition zones were measured in incubation at 35 °C. A calibration curve for the inhibition of the organism was also determined using the paper with loading standard concentration of each antibiotic agent (1, 10, 100, and 1000 μg/mL), respectively. The release concentration of antibiotic agent was then determined by interpreting the curve. The bioactivity of the antibiotic agent on organism (*S. aureus*) was calculated using the following equation:(1)$$\mathrm{Bioactivity}\;\left(\%\right)=\frac{diameter\;of\;sample\;inhibition\;zone}{diameter\;of\;maximum\;inhibition\;zone}$$

For the tests of optical densities of the bacterial suspension solutions with and without the antibiotic agent loaded on the samples, two concentrations of bacterial suspension solutions (10^6^ CFU/mL and 10^8^ CFU/mL) for the *S. aureus* were employed in order to estimate the antibacterial activity of the samples, respectively. Samples was kept at the solution (3 mL) at a concentration of 15 mg/mL for the ampicillin sodium salt or the vancomycin hydrochloride for two days, respectively. Standard test with only organisms in the solution was also carried out in order to estimate the growth rate of organisms as a function of time. The bacterial suspension solutions were kept at 37 °C with the shaking rate of 180 rpm. The values of optical density for the bacterial suspension solutions with and without the antibiotic agent on the samples were measured using the scanning spectrophotometer (Shimadzu, UV-1601PC, Tokyo, Japan) with the incident light wavelength of 600 nm. The relative value of optical density of the bacterial suspension solution was estimated using the following equation:(2)$$\mathrm{Relative}\;\mathrm{value}\;\mathrm{of}\;\mathrm{OD}\left(\%\right)=\frac{OD\;value\;of\;solution\;}{OD\;value\;of\;standard\;solution}$$

## 3. Results and Discussion

A pre-test with direct absorption of antibiotic agent onto the PEEK substrate was carried out in order to estimate the possibility for the direct loading of antibiotic agent onto the PEEK substrate. Appendix A shows the variations of concentrations for various antibiotic agents in the solutions containing the PEEK sample within 120 h. Almost constant concentrations for all antibiotic agents in the solution baths with/without shaking the samples were observed using the HPLC analysis. This indicates that the direct absorption of antibiotic agent onto the PEEK samples is difficult, and therefore we have to develop the new procedure for the loading of antibiotic agent onto the PEEK sample. Because the ZnO nanorod arrays may provide enough active surface areas for the direct absorption of these antibiotic agents, as shown in the introduction section, the growth of ZnO nanorod arrays on substrates were carried out. For the determinations of optimal parameters for ZnO nanorod arrays grown on PEEK sample, we used the glass substrate for the preparation of ZnO nanorod arrays, which was the same as those used in the literature [34]. Various concentrations (4.0–5.5 M) for ammonium hydroxide solutions with a total volume of 3 mL for the growth ZnO samples on substrates were carried out. Appendix A shows the XRD patterns of ZnO samples grown on the glass substrates after the thermal annealing process.

The XRD peaks for samples a–d agree well with those reported in the wurtzite ZnO phase (JCPDS card no. 89–510). With an increase in the concentration for ammonium hydroxide in the solution, the peak intensity at 2θ of 34.45° correspondeds to the increase in (0 0 2) crystal plane, which indicates that the samples were the wurtzute ZnO samples with the (0 0 2) prefer orientation. Although the XRD patterns of samples showed that the ZnO samples grown on glass substrates were polycrystalline ZnO samples with (0 0 2) crystal plane prefer orientation, it is necessary to examine the surface morphologies of ZnO samples using the SEM images. Appendix A shows the SEM images of samples a–d at 50 k(X). From the SEM images shown in Appendix A, the ZnO nanorod arrays on the glass substrates can be observed for all samples. With an increase in the ammonium hydroxide concentration in the reaction bath, the average diameter of the ZnO nanorods increased. The distributions of diameter of ZnO nanorods for samples a–d obtained from Appendix A are 80 ± 20 nm, 150 ± 30 nm, 180 ± 40 nm, and 400 ± 40 nm, respectively. Sample a has the smallest average diameter for ZnO nanorod and the largest number of nanorod arrays. Sample d has the largest average diameter for ZnO nanorod and the lowest number of nanorod arrays. The compositions of samples a–d are also analyzed using the EDS analysis. The [Zn]/[O] for samples a–d were a litter greater than 1, which indicated that the samples had minor oxygen vacancies. Sample a had the lowest [Zn]/[O] value of 1.06, while sample d had the highest [Zn]/[O] value of 1.26. The highest [Zn]/[O] value for sample d indicated that it may have lowest resistivity and can be used for the optical-electronic device [35,36]. However, for the applications of drug absorption and the control of the drug release rate using ZnO sample, the value of active surface area for sample is more important than its electrical property. Then, we used specific surface area analyzer to estimate their active surface areas. The active surface areas of samples a–d are around 4.3, 2.1, 1.4, and 0.6 m^2^/g, respectively. The active surface area of ZnO sample for the absorption decreases with an increase in the ammonium hydroxide concentration in the reaction solution. Sample a has the largest number of nanorod arrays, although it has the lowest diameter of the ZnO nanorod. The largest number of ZnO nanorod arrays on substrate indicates that it has the largest active surface area. Because sample a has the largest active surface area, it seems a good drug carrier for the absorption of antibiotic agent. Hence, we used the sample a (the concentration of ammonium hydroxide of 4 M in precursor solution for the growth of ZnO sample) as the antibiotic agent carrier for the loading of various amounts of antibiotic agents with the direct absorption of antibiotic agents in solution bath. Figure 1 shows the XRD patterns of PEEK samples before and after the ZnO nanorods growth with the concentration of ammonium hydroxide kept at 4 M (sample a) in precursor solution. The XRD pattern of sample indicated that the ZnO can be grown onto the PEEK substrate using the simple chemical synthesis. The inset figures in the Figure 1 show the pictures of the PEEK samples before and after ZnO growth. They show a white film covered at the PEEK sample surface. The XRD pattern of sample shows that the crystal phase of ZnO sample agrees well with those reported in the JCPDS card (no. 89–510), which indicates the sample is the polycrystalline hexagonal ZnO phase. However, the (0 0 2) crystal plane preference orientation for ZnO sample on PEEK sample is not observed, which may be due to surface property of sample. It is well known that the surface properties of substrates influence the crystal phases of metal oxides grown on substrates [37]. However, an effective seed layer can contribute the formation of ZnO nanorods on the substrate in the alkaline solution, although the XRD pattern of sample shows the polycrystalline hexagonal ZnO phase without (0 0 2) crystal plane prefer orientation [34]. Without the seeding of nucleation centers (Mn or Fe ions) onto non-conductive glass substrates, little or no ZnO film can be deposited onto the glass or plastic substrates [34]. Seed layers on substrates can also promote columnar growth (prefer orientation growth), although the polycrystalline ZnO phase was observed in the XRD patterns of samples [34]. Figure 2 shows the FE-SEM images of ZnO nanorods grown on PEEK substrate with the concentration of ammonium hydroxide kept at 4 M. Figure 2a,b show the SEM images of ZnO/PEEK sample using simple chemical synthesis at 10 and 50 k(X), respectively. From the SEM image of sample, it can be confirmed that the morphology of ZnO grown on PEEK substrate is the nano- to submirco- rod arrays. The diameter distributions of ZnO sample grown on PEEK substrate are in the range of 100–400 nm, which is a little larger than the ZnO sample grown on the glass substrate. The specific surface area analyzer was also employed to estimate active surface area of ZnO grown on PEEK substrate. The active surface area of ZnO/PEEK sample is around 4.0 m^2^/g, which is a little lower than that prepared at the glass substrate with the same procedure.

Then, we used the ZnO/PEEK samples to absorb antibiotic agents with various concentrations in the solution baths. The loading percentage for the antibiotic agent on substrates was calculated using the difference of concentration for antibiotic agent in the solution bath before and after absorption. Detail testing parameters are given in the Table 1. From the results shown in Table 1, total absorption amounts of ampicillin loaded on the samples (samples (A–C)) were lower than those for vancomycin loaded on the substrates. This is possibly due to the higher numbers of hydroxyl groups in the vancomycin compared with those in the ampicillin, which can increase the loading amount of vancomycin onto the ZnO sample. High loading amount of antibiotic agent on the sample may increase its long-term antibacterial property with the chemical bonding and therefore show better clinical performance for musculoskeletal wounds.

Figure 3 shows the release profiles and the cumulative amounts of ampicillin salts from the ZnO samples into the buffer solution as a function of time. Almost stable concentrations of ampicillins released from the samples (A–C) in the buffer solution can be observed; they maintain the value of higher than the minimum inhibition concentration (MIC) 90 for *S. aureus* in 96 h [29]. The insert figure in Figure 3a shows a fast decrease in the concentration of ampicillin in the buffer solution at around first 2 h after the beginning of drug release test. The possible reason for the decrease in concentration of ampicillin in buffer solution may be due to the weak absorption of ampicillin at the ZnO nanorod array surfaces. When the samples contacted with the buffer solution, the ampicillin with weak absorption on the ZnO surface released into the buffer solution due to the mass transferring driving force. The mass transferring driving force for ampicillin is a function of the difference of the concentrations between the sample surface and that in the buffer solution. The concentration of ampicillin in the buffer solution increases to the highest value at the first hour due to its highest weak absorption amount of ampicillin at ZnO surface (low number of hydroxyl groups). Please note, a buffer solution was refreshed every hour. The decrease in the concentration of ampicillin in the buffer solution at the second hour may be due to the ampicillin running out with the weak absorption on the ZnO sample. It can be observed that the concentrations of ampicillin in the buffer solution increase in the period of 2–4 h, because ampicillin begins to release from the ZnO samples into buffer solution with the chemical absorption on samples. Stable drug release concentration in the buffer solution from the ZnO sample can observed in the period of 8–96 h for all samples, and they are higher than the value of MIC 90 for the *S. aureus*. Figure 3b shows the cumulative amounts of ampicillin from the samples into buffer solution as a function of time. Sample (A) has the highest release percentage of ampicillin from sample into buffer solution, while that for sample (C) has the lowest percentage. Low cumulative amount of ampicillin into the buffer solution indicates that the most of surface areas for ZnO sample are still covered with the antibiotic agents, and therefore the fast release of Zn^2+^ ions from sample into the solution may be inhibited [18]. In order to obtain the evidence of release amounts of Zn^2+^ions from our samples into the buffer solution, the concentration of Zn^2+^ ions in the buffer solution was analyzed using the ICP-OPS. The results showed that the concentration of Zn^2+^ions from the sample (C) approached 0 ppm after the 96 h test, while that from the pure ZnO nanorod arrays samples was around 1 ppm after 96 h test. Compared with the Zn^2+^ concentration of 6 ppm in the solution reported by Xiang et al. [18], our results showed very low concentration of Zn^2+^ ions released from both ZnO and ZnO/antibiotic agents samples. This is possibly due to the annealing process and the low solubility product of (k_sp_ = 6.8 × 10^−17^) for the ZnO samples. The low solubility of ZnO in buffer solution may result in the low release rate of Zn^2+^ ions from sample, and therefore the concentration of Zn^2+^ is much lower than 6 ppm, which was reported to be cytotoxic in vitro [18,38]. The low concentration of Zn^2+^ ions in the solution bath can decrease in terms of cell toxicity.

Figure 4a,b also shows the release and cumulative profiles of the vancomycin from the ZnO samples into the buffer solutions, respectively. The release profiles for the vancomycin from the ZnO samples into buffer solution are similar to those for ampicillin released from the ZnO samples. The variations of release profiles for the vancomycin from the ZnO samples into buffer solution were larger than those for the ampicillin. However, the concentrations of vancomycin in the buffer solutions are still higher than the value of MIC 90 for vancomycin on *S. aureus* [29]. Large variation of concentration for vancomycin in the buffer solutions may be due the physical property of vancomycin. The pH value of solution containing vancomycin is lower than that for ampicillin due to high numbers of acidic function groups for vancomycin (pK_a_ for vancomycin of around 2.99 and that for ampicillin of around 3.24) [39]. Low pH values (pH value < 4 in our study) in buffer solution can be observed near the sample. Therefore, the following reaction would take place at the sample surface [18].
ZnO + 2H^+^ → Zn^2+^ + H_2_O(3)

The decomposition of ZnO samples took place with the reaction of H^+^ ions released from antibiotic agent and resulted in the release of antibiotic agent and Zn^2+^ ion from ZnO sample surface into the buffer solution. Please note, due to the lower pH value near the vancomycin/ZnO sample, the larger variation of concentration of vancomycin in the buffer solution was observed compared with that for ampicillin in the buffer solution. However, the results from the ICP-OES showed that the concentration of Zn^2+^ ions in the buffer solution approached zero, both for the ampicillin and vancomycin/ZnO nanorod/PEEK samples. In order to find out the reactions taking place at the ZnO/antibiotic agent (vancomycin) samples, the SEM images of ZnO/vancomycin/PEEK samples were carried out.

Figure 5 shows the SEM images of sample (F) before and after drug release test. SEM image of pure ZnO nanorod array sample after test was also employed for comparison. SEM image of sample (F) showed that the decomposition of ZnO nanorod arrays took place after the drug release test. Pure ZnO sample without the loading of antibiotic agent showed the similar nanorod morphology, although few decompositions of ZnO nanorod arrays were observed. This could explain why few Zn^2+^ ions were detected in the buffer solution using ICP-OES analysis. The SEM images shown in Figure 5 confirmed that the decomposition of ZnO nanorod arrays took place during the drug release test. However, the ICP-OES results showed that small amounts of Zn^2+^ions diffused into the buffer solution. This is possibly due to the low solubility of ZnO material in the nearly neutral solution [18,38]. Decomposition of ZnO sample took place with the H^+^ ions released from the antibiotic agent and caused the release of Zn^2+^ions from the ZnO sample into the buffer solution. Kokotov and Hodes [34] reported that the existence of hydroxyl group on substrates is the key factor in the reaction with the metal ions such as Zn^2+^ ions. Many hydroxyl groups were formed at ZnO sample surface when they were contacted with buffer solution. These Zn^2+^ions in solution then reacted with the hydroxyl groups on the sample surface to form the uniform ZnO thin films on PEEK substrate, as shown in Figure 5.

For the evaluations of drug release mechanisms for the antibiotic agents released from the samples, four possible release mechanisms are proposed to fit the experimental data, which are the (I) zero order, (II) first order, (III) Higuch, and (IV) Koresmeyer–Pappas kinetic models. Detailed descriptions can be found in our previous study [29]. Detailed fitting results are shown in Appendix A. The Koresmeyer–Pappas model can give the best fitting results. Detailed parameters of the Koresmeyer–Pappas model for samples (A–F) are given in Table 1. The rate constants of antibiotic agents from the ZnO samples are in the range of 4.4–9.2 h^−1^ and decrease with an increase in the concentration of antibiotic agent in the solution bath. Fast rate constant indicates the fast drug release rate from sample into the buffer solution. The drug release profiles of ampicillin and vancomycin from the samples into the buffer solution indicated that the antibacterial properties of ZnO samples may continue for at least 96 h in the test (for all concentrations of antibiotic agents released from samples of higher than the value of MIC 90 for antibiotic agent on *S. aureus* in buffer solutions). In order to confirm real antibacterial properties of ZnO samples, the bioactivity of *S. aureus* with samples (C) and (F) was also tested using the disk diffusion method. Figure 6a,b shows the bioactivity and the estimation of drug release distributions of ampicillin and vancomycin in the Petri disks as a function of time using the calibration curve for the inhibition of *S. aureus*, respectively.

From the results shown in Figure 6; the bioactivity of the ampicillin/ZnO sample on the *S. aureus* was kept at 100% at the first 50 h and decreased to around 35% after 360 h (around 15 days) test. Low concentration of ampicillin released from sample (C) in the Petri disk (corresponding to the paper with loading antibiotic agent concentration of around 1 μg/mL in solution bath) was the major factor in its poor long-term bioactivity. The bioactivity of vancomycin/ZnO sample showed the good antibacterial properties of *S. aureus* due to its high concentration of vancomycin released from sample (F) (corresponding to the paper with loading vancomycin of 1000 μg/mL in the solution bath). The bioactivity of sample (F) maintained around 100% in the first 50 h. After that, the bioactivity of sample (F) decreased due to the decrease in concentration of vancomycin released from the sample. However, the bioactivity of sample (F) still maintained at least 80% in the 650 h test (28 days). For the long-term bioactivity test, bioactivity of ZnO/vancomycin sample was higher than that for the ZnO/ampicillin sample. The measurements for optical densities of the *S. aureus* in the solutions with and without antibiotic agent loading onto the substrate were also employed for the evaluation of inhabitation behavior of the bacterial growth [40,41,42]. When the value of optical density in the solution containing organisms increased, the amounts of organisms in the solution increased, while with the decrease in the value of optical density, the growth of organisms in the solution was reduced. Figure 7 shows the relative optical density (OD) values for the *S. aureus* in the solution containing the ZnO samples without and with the absorption of antibiotic agents (samples (C) and (F)). The OD value of *S. aureus* in the same solution as a function of time was used as the standard test. For the high concentration of *S. aureus* in the solution (Figure 6a), the growth rate of *S. aureus* in the solution was inhibited with the ZnO/ampicillin or ZnO/vancomycin samples in the solution. Only ZnO sample showed the relatively poor antibacterial properties of the OD value in the solution. The antibacterial property of pure ZnO sample is due to its nanorod arrays, which can rupture the *S. aureus* membrane when the cell attaches to these microstructures. However, the OD value approached constant after the 4 h test. This may be due to the surface of ZnO sample covered with the dead *S. aureus*, which resulted in the poor antibacterial property of pure ZnO samples after several hours. Similar result was also reported by Singh et al. [28]. For the tests of samples with antibiotic agents loaded on the ZnO samples, sample (C) can inhibit the growth of *S. aureus* with high concentration after one hour test. It confirmed that the antibacterial property of ZnO/ampicillin sample is better than that for the ZnO/vancomycin sample in initial situation, although the better long-term antibacterial property of ZnO/vancomycin sample is observed, as shown in Figure 6. At the lower concentration of organism in the solution (Figure 6b), the relative OD values can be reduced to around 30% in 6 h using the ZnO/ampicillin sample. These results show the direct absorption of antibiotic agent onto the ZnO nanorod arrays sample has good antibacterial properties. The mixture of ampicillin/vancomycin in the solution bath with the suitable ratio for absorption onto the PEEK samples will be developed for biomaterials with good short- and long-term antibacterial properties. It will be done in the near future.

## 4. Conclusions

In this study, we developed a simple and low-cost method for the preparation of ZnO nanorod arrays on the PEEK substrate for possible application in biomaterials. When the concentration of ammonium hydroxide in the precursor solution was increased, the average diameter of ZnO nanorods increased but the total number of the ZnO nanorod arrays decreased, which resulted in different active surface areas of ZnO samples. When the concentration of ammonium hydroxide was kept at 4.0 M in the precursor solution, the maximum active surface area of ZnO nanorod sample was obtained. Various loading amounts of antibiotic agents for the ZnO/ampicillin or ZnO vancomycin samples can be obtained using direct absorption in solutions containing antibiotic agents. With the increase in the concentration of antibiotic agents in the aqueous solution, the amount of antibiotic agents absorbed onto the sample surface also increased. Stable drug release profiles for the ampicillin and vancomycin from the samples into the buffer solution were observed and produced the antibiotic agent concentrations in buffer solutions higher than those of MIC 90 on the *S. aureus* within 96 h. The drug release kinetics for the antibiotic agents released from the samples agreed well with the Korsmeyer–Peppas model. Bioactivities of the ampicillin and vancomycin loaded on the ZnO/PEEK substrates were maintained for at least 15 and 28 days, respectively. Relative optical density of *S. aureus* in the solution with the concentration 10^6^ CFU/mL can inhibit the growth of *S. aureus* at around 70% using the ZnO/PEEK sample in six hours. This study showed that a simple and low-cost method for the preparation of ZnO/antibiotic agent materials could have further applications in biomedical-related technology.

## Figures and Tables

**Figure 1 nanomaterials-09-00713-f001:**
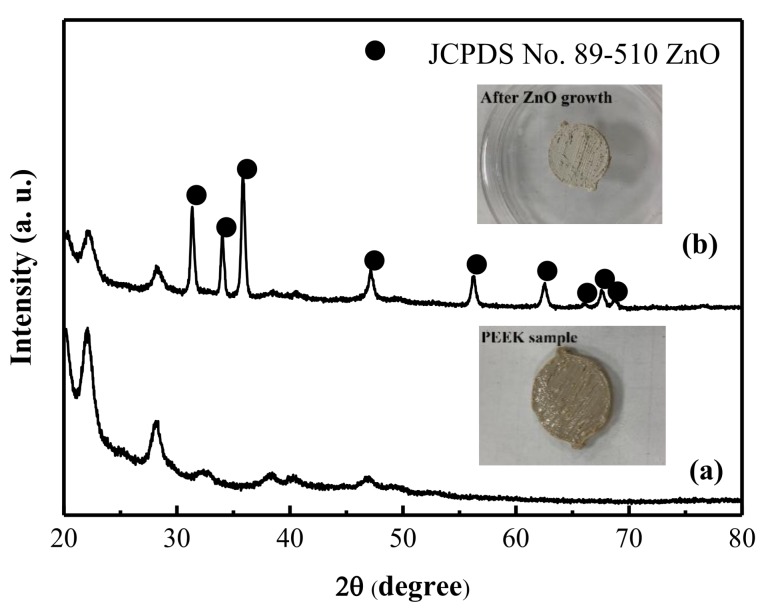
XRD patterns of (**a**) polyetheretherketone (PEEK) sample and (**b**) ZnO nanorods/PEEK sample.

**Figure 2 nanomaterials-09-00713-f002:**
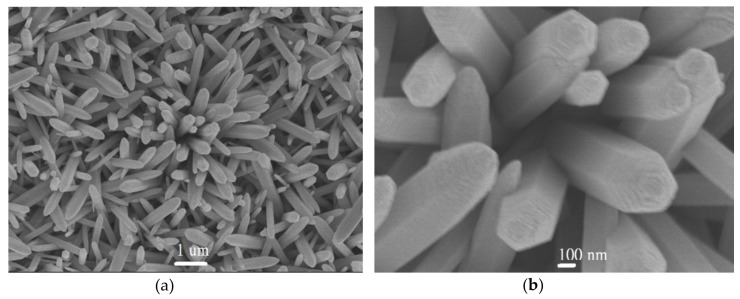
SEM images of ZnO nanorods/PEEK sample at (**a**) 10 k(X) and (**b**) 50 k(X), respectively.

**Figure 3 nanomaterials-09-00713-f003:**
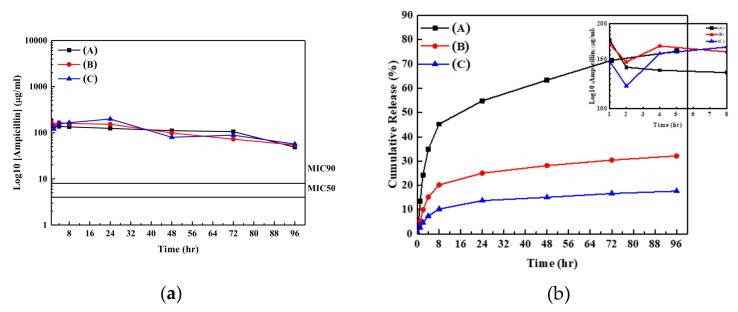
(**a**) Release profiles and (**b**) the cumulative amounts of ampicillins released from samples (A–C) into the buffer solution as a function of time.

**Figure 4 nanomaterials-09-00713-f004:**
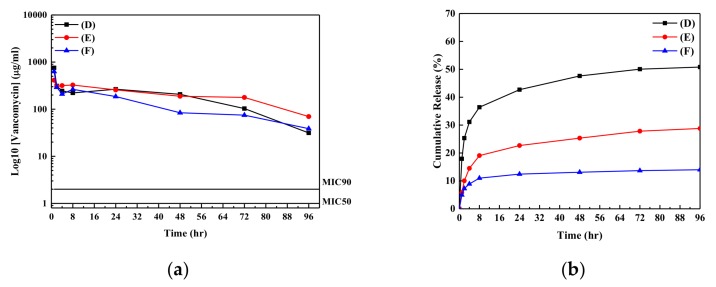
(**a**) Release profiles and (**b**) the cumulative amounts of vancomycin released from samples (D–F) into the buffer solution as a function of time.

**Figure 5 nanomaterials-09-00713-f005:**
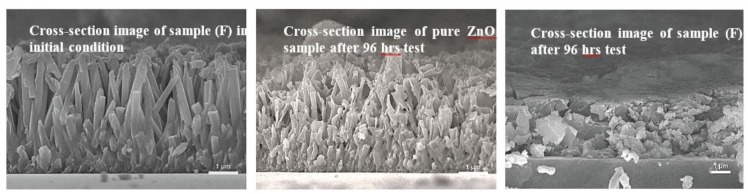
SEM images of pure ZnO sample and sample (F) before and after drug release testing, respectively.

**Figure 6 nanomaterials-09-00713-f006:**
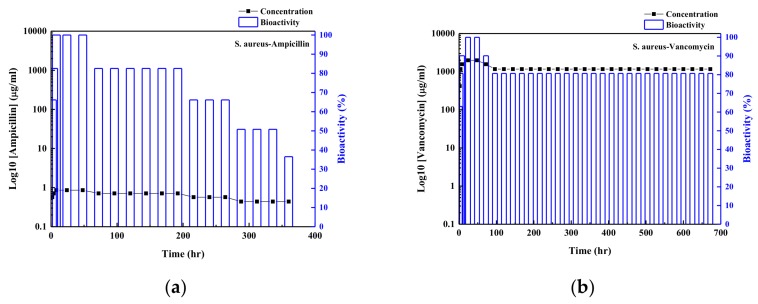
Bioactivities for *S. aureus* using the release of (**a**) ampicillin (sample (C)) and (**b**) vancomycin (sample (F)) from the antibiotic agent/ZnO nanorod arrays samples, respectively.

**Figure 7 nanomaterials-09-00713-f007:**
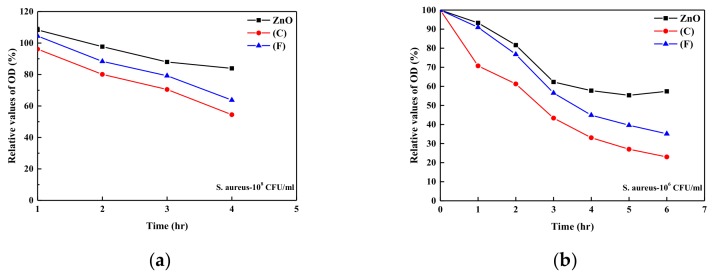
Relative optical density values as a function of time for (**a**) the initial *S. aureus* concentration of 10^8^ CFU/mL and (**b**) 10^6^ CFU/mL with pure ZnO sample; samples (C) and (F) in the solutions.

**Table 1 nanomaterials-09-00713-t001:** The absorption percentages and parameters for drug release profiles with various antibiotic agents used in this study.

Sample	Antibiotic Agent for Absorption	Concentration of Antibiotic Agent for Absorption	Absorption Percentage (%)	Rate Constant for K-P Model (hr^−1^)	*n* Values for the K-P Model	Value of R^2^
(A)	Ampicillin	5 mg/mL	17.4	9.05	0.27	0.97
(B)	Ampicillin	10 mg/mL	21.1	6.58	0.28	0.95
(C)	Ampicillin	15 mg/mL	25.6	4.35	0.32	0.94
(D)	Vancomycin	5 mg/mL	56.1	9.22	0.17	0.98
(E)	Vancomycin	10 mg/mL	47.5	6.61	0.25	0.94
(F)	Vancomycin	15 mg/mL	56.4	5.68	0.16	0.93

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
