# Peer review of "Antibacterial Application on Staphylococcus aureus Using Antibiotic Agent/Zinc Oxide Nanorod Arrays/Polyethylethylketone Composite Samples"

_nanomaterials, 2019, doi:10.3390/nano9050713_

Round 1

Reviewer 1 Report

The manuscript by Dave W, Chen et al pertaining the development of PEEK-based materials coated with ZnO nanorods as antibacterial materials in wound treatment is of significant scientific quality. However, serious presentation issues make it non-suitable for publication in its present form. There are several points that need to be addressed so that the manuscript can be understandable, properly-structured and coherently-written. In addition, there is a serious reservations about some of the experimental data presented (see points 9 and 10). Part of the reluctance to accept could be due to the language issues that make the text hard to understand.

The manuscript requries extensive language editing. It is suggested that it is proof-read by a native speaker before resubmission. There are several parts of the text that contain grammatical and syntax mistakes and other parts that the text is very hard to read and comprehend.

The abstract contains too many technical details and does not allow the reader to understand what the manuscript is all about. It must be rewritten in its entirety in order to demonstrate the novelty of the work, its highlights and its general conclusions.

The introduction is disproportionately lengthy and unnecessarily wordy. The presentation of the sate-of-the-art and the related literature resembles a review article and is not suited for a an article. It is recommended to reduce this part (lines 43-84 and lines 95-101) and keep only the information that is mostly relevant to the manuscript's content (e.g. the technical details of various methods of ZnO synthesis are superfluous). In addition, despite the length introduction,  the novelty of the work and the importance of choosing PEEK are not stressed enough.

A short description of the 3d-printing method of PEEK would be nice. The bare referral to [29] and complete lack of description is in stark contrast to the superfluous details provided for the work of other references in the introduction.

Why did the authors choose the seeding layer formation method of [31]? Why does it have advantages compared to other sol-gel techniques that use simpler solvents (e.g. water or alcohols) and Zn-based precursors? Is it better suited for PEEK? It seems oddly complex with repsect to other reports in literature.

Why were the samples annealed? What does RTA offer with respect to simple heating (process escribed in lines 140-142).

Section 3 contains redundant text of things already described in Section 2 (e.g. lines 213-217 and 275-278). Repetition should be taken care of.

Lines 219-222 refer to the methods used and should be aprt of Section 2.

Samples solvothermally/hydrothermally grown on different substrates should not be compared or taken fro granted that are similar. It is established in literature that the morphology of the nansotructures strongly depends on the mature of the substrate (see for example Procedia Engineering 120 ( 2015 ) 447 – 450).  ZnO nanorods grown on glass substrates should not be assumed to be the same as the ones grown on PEEK, hence the analysis based on those cannot and should not be used (Fig.1 and related discussion).

10. it is not clear in the text whether the SEM images correspond to the nanoroods grown on PEEK or on glass. If they are grown on PEEK, then its is OK to present them and use the morphological features in the analysis. If not, then the subsequent analysis is problematic since it is not guaranteed that the nanorods on PEEK share the same features and hence the same functionalities.

Which sample was chosen to be studied with the antibiotics (Table 1) and the subsequent experiments with S. aureus. It is not clear in the text.

Comparison between the XRD data of Fig.1 and Fig.5 as well as a more detailed analysis are missing.

The resutl analysis between lines 399-437 is very hard to follow -mostly because of the language problems.

Author Response

Ms. Ref. No.: nanomaterials-475831

Answers to Reviewer’s comments  

We are grateful for your comments on the manuscript. Your comments have been carefully examined and replied as following. All modified parts have been clearly marked in this revised manuscript

Reply to Reviewers' comments:

The reply of every question is attached as pdf file

Reviewer 2 Report

1. What sense does it make to provide the specific surface area value to the 4th decimal place in the abstract (18) ?  Do the authors achieve the specific surface area measurement accuracy with an error to the 4th decimal place?

2. Development of methods of obtaining ZnO nanostructures with controlled properties, such as shape or size, enables their repeatable application, e.g. as antibiotic carriers. At present, the synthesis of nanomaterials is at a new stage of development, where it is a common requirement to control the properties of the obtained nanostructures, their repeatability and reproducibility. I encourage the authors to underline this fact in the “Introduction” and add a sentence of a comment about the development of such methods of obtaining nanostructures, from nanoparticles to nanorods, with the following publications serving as examples:

- Size control mechanism of ZnO nanoparticles obtained in microwave solvothermal synthesis, 2018, Nanotechnology, 29, 065601

- Morphology and size controlled synthesis of zinc oxide nanostructures and their optical properties, J Mater Sci: Mater Electron, 2018, 29, 9339.

- Size Control of ZnO Nanorods Using the Hydrothermal Method in Conjunction with Substrate Rotation, Journal of Nanoscience and Nanotechnology, Volume 17, Number 11, November 2017, pp. 7952-7956(5)

3. There is no model and name of the thermal annealing system manufacturer (142).

4. There is no description of preparation of samples for specific surface area measurement (148). Please explain the sense of the parameter “specific surface area micomeritics”. Please put the text in order, now the “Results” part contains information about the parameters of the specific surface area analysis (255-257). This information should be included in the “Materials and Methods” part. Was a “new method” used for measuring the specific surface area? The description of the measurement performance is quite surprising to me. Why was nitrogen held at the temperature of 90°C?

5. I have noticed the lack of logical coherence of the result presentation in the text; namely the authors describe the sense of their research in “Results and discussion” (199), while the goal of the research must be placed at the end of “Introduction”. The successive two sentences (199-201) concern procedures, while this information must be included in “Materials and methods”. Later, the authors inform about what is included in the “supporting file”, without discussing these results. And so on … The text must be put in order and corrected in accordance with the general guidelines of “Nanomaterials”.

6. Does the growth of ZnO nanorods occur identically on glass substrates and PEEK sample? Did the nanorods obtained with the same synthesis parameters but on different substrates have the same properties? If yes, where is the proof in the form of results? (224-225) and (239)

7. Please calculate approximate diameters of the nanorods from the SEM photographs (246-249).

8. EDAX acronym is wrong, you should use EDS acronym in accordance with ISO 22309:2011 (147, 250). There is no information as to how many repetitions there were per sample, and how the sample was prepared for the EDS measurement.

8. What were the sizes of the PEEK/ZnO samples that were used for testing?

9. There is no information about the quantities of ZnO (mg/cm2) obtained in the form of ZnO on the surface of the PEEK samples.

10. The equation of reaction no. 3 is not correct (335).

Author Response

Ms. Ref. No.: nanomaterials-475831

Answers to Reviewer’s comments  

We are grateful for your comments on the manuscript. Your comments have been carefully examined and replied as following. All modified parts have been clearly marked in this revised manuscript. Please find them in the attached file

Round 2

Reviewer 1 Report

Thank you for the immediate response and the carefully made corrections.

Author Response

Referee: 1 Thank you for the immediate response and the carefully made corrections.

Ans.: We thank your comment on this revised manuscript.

Reviewer 2 Report

1. The high specific surface area of the samples received by the authors is quite debatable. Other authors received a nanorods with a much larger area (“Synthesis and Characterization of ZnO Nanorods Based on a New Gel Pyrolysis Method,” Journal of Nanomaterials, vol. 2011, Article ID 628203, 11 pages, 2011. https://doi.org/10.1155/2011/628203). I suggest removing the words "High Surface Area" in the title. Moreover in the abstract, information about the specific surface area has been removed.

2. The description of the surface analysis (“The gas for the analysis of sample’s surface area is the nitrogen gas with the pressure of set at 3 mmHg. Because the measurement of specific surface area for samples is calculated using the amount of nitrogen gas absorbed at sample surface, the sample has to be kept at the temperature of 90°C in order to avoid the influence of water content in the samples.”) it is unacceptable.
Please provide the following information:
- how the samples were prepared before the measurement (temperature, time, gas environment)
- type of method (was it the BET method ? (nitrogen adsorption method based on the linear form of the BET (Brunauer-Emmett-Teller) isotherm equation))
- used the adsorptive range P/Po.

It is suggested to read for example: https://www.usp.org/sites/default/files/usp/document/harmonization/gen-chapter/g11_pf_30_4_2004.pdf

3. Was the phosphate buffer solution (PBS) made or bought (171-172) ?

Author Response

Reply to Reviewers' comments:

Referee 2:

1. The high specific surface area of the samples received by the authors is quite debatable. Other authors received a nanorods with a much larger area (“Synthesis and Characterization of ZnO Nanorods Based on a New Gel Pyrolysis Method,” Journal of Nanomaterials, vol. 2011, Article ID 628203, 11 pages, 2011. https://doi.org/10.1155/2011/628203). I suggest removing the words "High Surface Area" in the title. Moreover in the abstract, information about the specific surface area has been removed.

Ans.: We thank your important comment on our revised manuscript. According to your comment, we delete the “High Surface Area” in the title. The title of this revised manuscript become “Antibacterial Applications on Staphylococcus aureus Using Antibiotic Agent/Zinc Oxide Nanorod Arrays/Polyethylethylketone Composite Samples

2. The description of the surface analysis (“The gas for the analysis of sample’s surface area is the nitrogen gas with the pressure of set at 3 mmHg. Because the measurement of specific surface area for samples is calculated using the amount of nitrogen gas absorbed at sample surface, the sample has to be kept at the temperature of 90°C in order to avoid the influence of water content in the samples.”) it is unacceptable.
Please provide the following information:
- how the samples were prepared before the measurement (temperature, time, gas environment)
- type of method (was it the BET method ? (nitrogen adsorption method based on the linear form of the BET (Brunauer-Emmett-Teller) isotherm equation))
- used the adsorptive range P/Po.

Ans.: We thank your important comment on our revised manuscript. According to your comment, we have added the detail procedure for the sample preparation, type of measurement and the adsorptive range P/Po in this revised manuscript. Please find them in line 152-162 in the revised manuscript.

For the measurement of specific surface area of a sample, a degassing process has to be carried out in order to remove the gases that may have physically absorbed onto the sample surface. For the degassing process, the sample (of around 0.3 g) was put in a container loaded in the specific surface analyzer. The temperature and pressure of the container were kept at 90°C and 3mm-Hg in order to remove the gases physically absorbed onto the sample, respectively. Total degassing process time was kept at 1000 min in order to obtain a clean sample. After the degassing process, total weight of the clean sample without any gas absorbed on the sample can be obtained using the weighting method. The gas for the analysis of sample’s surface area is the nitrogen gas with the temperature kept at 77.4 K (the boiling point of nitrogen). The special surface area of sample was calculated using the nitrogen adsorption method based on the linear form of the BET (Brunauer-Emmett-Teller) isotherm equation with the absorption range of P/Po kept in 0.06-0.6.

3. Was the phosphate buffer solution (PBS) made or bought (171-172) ?

Ans.: We really thank your important comment. We bought the phosphate buffer solution from the Sigma-Aldrich Co. It was added in the line 178 in the revised manuscript.
